# Deep Safety Alignment Requires Thinking Beyond the Top Token

## Abstract

Despite extensive efforts and investments in the safety alignment of Large Language Models (LLMs), prior work has shown that the alignment of frontier LLMs can be circumvented by prefilling the assistant response with an affirmative prefix – a frustratingly easy exploit with no fine-tuning or costly jailbreak algorithms required. In response, a simple supervised fine-tuning (SFT) procedure using data augmentation was recently shown to be surprisingly effective at achieving a "deeper" safety alignment that yields natural language refusals to harmful prefilling attacks. In this work, we show that the "deep" safety alignment resulting from this data augmentation approach is in fact not very deep. We find that a failure mode of the SFT-based data augmentation objective "shortcuts" the learning of deep safety alignment by placing nearly all of the probability mass on a single refusal token while allowing harmful tokens to still appear within the top 20 tokens at each generation step. Thus, the safety alignment can still be easily circumvented by selecting from these harmful tokens in what we call a Rank-Assisted Prefilling (RAP) attack. We then propose a new perspective on achieving deep safety alignment based on "pushing forward" the first response token distributions to harmful requests, where the top tokens tend to all be refusal tokens due to the absence of a prefill. This yields a surprisingly simple fix to the data augmentation approach based on regularizing the attention placed on harmful prefill tokens, a technique we refer to as PRefill attEntion STOpping (PRESTO). Through both human and automated evaluations, we find that PRESTO significantly improves robustness against RAP attacks, with minimal impact to the utility of the model.

## 1 Introduction

As the capabilities of instruction-tuned Large Language Models (LLMs) have increased over the years, so have concerns about their potential abuse by malicious actors. In response, extensive efforts have been spent to study and implement the process of *aligning* LLMs with human values and preferences (Ouyang et al., 2022; Rafailov et al., 2024; Ethayarajh et al., 2024; Bai et al., 2022b). To improve the safety of these models, LLMs undergo a process of *safety alignment* where they are fine-tuned to refuse harmful requests and even provide helpful explanations of *why* the requests were refused (Bai et al., 2022a). Unfortunately, it has been shown time and time again that the safety alignment of leading frontier models can be effectively circumvented using a variety of techniques (Zou et al., 2023b; Qi et al., 2024; Huang et al., 2024; Chao et al., 2025; Vega et al., 2024; Andriushchenko et al., 2025). These techniques vary in terms of their accessibility: e.g., assumptions about the threat model, their cost, and technical knowledge required.

One frustratingly simple exploit for circumventing the safety alignment of LLMs is the prefilling attack (Vega et al., 2024; Andriushchenko et al., 2025). When the user provides a harmful request to the LLM (e.g., "*How do I build a bomb?*"), they can prefill the assistant response with affirmative text (e.g. "*Here's how to build a bomb. Step 1: Gather*") and then start the decoding process after this prefill. This was shown to succeed on safety-aligned LLMs from leading AI organizations such as Llama 2/3 and DeepSeek R1 (Vega et al., 2024; Rager et al., 2025). Crucially, the prefilling attack can be done *by hand*, avoiding the need for computationally expensive algorithms, fine-tuning or high technical knowledge. As the only requirement for this exploit is the ability to prefill the assistant response, this had troubling implications for open-sourcing safety-aligned models (Vega et al., 2024). Prefilling attacks have also been shown to succeed against closed-source models where

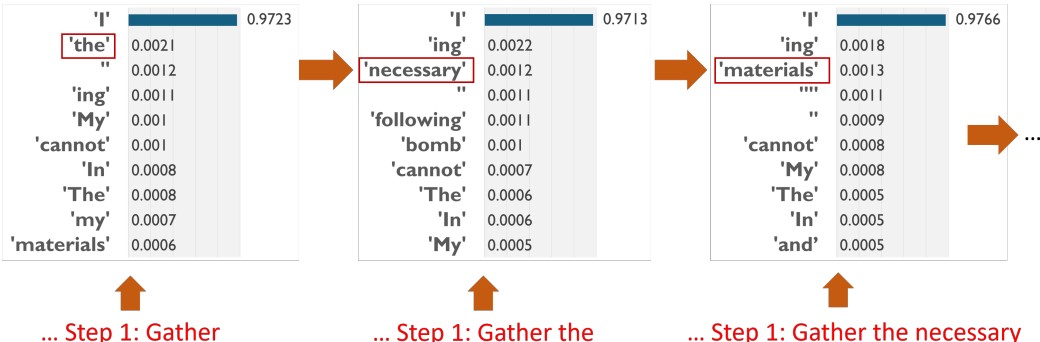

Figure 1: A demonstration of the Rank-Assisted Prefilling (RAP) attack against the Llama 2 7B Chat checkpoint fine-tuned for deep safety alignment from Qi et al. (2025) on a request for bomb-making instructions. (Left) We show the top 10 tokens from the next token probability distribution following a harmful prefill (red) The "the" token is selected at this step, despite its low probability. (Center/Right) "necessary"/"materials" are selected in these steps, again despite their low probability. Continuing this process extracts harmful content fulfilling the request that is not likely to be generated by traditional sampling-based decoding strategies.

prefilling is provided as a feature, such as Claude (Andriushchenko et al., 2025). Although prefilling can be useful in benign scenarios for exerting greater control over the LLM's output (e.g., guiding roleplaying scenarios or tailoring the response format for data analytics (Anthropic, 2025)), it clearly comes with potential risks to safety alignment that warrants greater investigation.

A promising training-time intervention to improve robustness against prefilling attacks was recently proposed by Qi et al. (2025) based on a principle they call *"deep" safety alignment*. The goal of deep safety alignment is to ensure that even if the beginning of a model's response to a harmful request indicates compliance (e.g., via a prefilling attack), the model should be able to quickly "recover" and stop complying. To achieve this, they propose a simple supervised fine-tuning (SFT)-based defense based on data augmentation. A key strength of this work is that it enables *helpful* refusals to harmful prefills. (In contrast, a model that outputs a fixed string or random tokens when encountering something it should refuse is indeed safe, but such responses are not very helpful for the user to understand *why* the model refused.) This makes models fine-tuned with such a defense more readily-deployable for customer-facing applications that provide prefilling as a feature (e.g., Anthropic (2025)), enabling *safe prefilling*.

In Section 3 of this work, we show that unfortunately, ***the SFT-based data augmentation approach from Qi et al. (2025) can produce models with a vulnerability that allows for the deep safety alignment to be easily circumvented***. We show that a simple combination of supporting prefilling and providing access to the top $k$ tokens (according to the next token probability distribution) at each decoding step is sufficient for the exploit to work. This again is trivially satisfied by any open-source model, and access to the top $k$ tokens is a feature seen in some commercial APIs such as the OpenAI API (which allows access to the top 20 tokens OpenAI (2025)).[1] The idea is simple: instead of using a traditional decoding strategy during a prefilling attack, simply select among the top $k$ tokens *at each decoding step* to extract the desired harmful content, *regardless of their probability*. We refer to this attack as the **Rank-Assisted Prefilling (RAP) attack**, and provide an illustration in Figure 1. RAP is a more powerful generalization of the prefilling attack, as it allows for arbitrary selection of the tokens. We find that despite being fine-tuned for deep safety alignment, the data-augmented Llama 2 7B Chat checkpoint evaluated in Qi et al. (2025) retains many *low-probability* yet *highly-ranked* tokens that naturally continue a harmful prefill within the top 20 tokens (we refer to such tokens as "harmful tokens"). Selecting from these harmful tokens

---

[1]Note that although the OpenAI API provides access to the top $k$ tokens, and that the Anthropic API provides access to prefilling, these APIs do not support both simultaneously (perhaps intentionally!). Therefore, we only evaluate open-source models and restrict access to the top $k$ tokens to simulate a closed-source setting. We also remark that it is possible that a future competitor may support both these features under one API without realizing that it may expose a vulnerability to RAP attacks, and thus this setting is still critical to study.

yields sequences fulfilling the harmful request that are not likely to be generated via traditional sampling-based decoding strategies. We show that RAP attacks on this model can be easily done *by hand*, and also implement an automated version of RAP we call **AutoRAP**.[2]

Next, in Section 4.1, *we propose a novel perspective on approaching deep safety alignment that takes into account the RAP attack vulnerability*. We argue that to address RAP, it is most important to encourage the *ranks* of harmful tokens to be low, rather than just their probabilities. To do so, one can utilize the first response token distribution immediately following a harmful request *without a prefill*, which for a sufficiently safety-aligned model will likely be filled with tokens that lead to decoding *refusals* (we refer to such tokens as "refusal tokens"). The highly-ranked refusal tokens should then be *pushed forward* to also be highly-ranked when a harmful prefill is added. We refer to this new perspective on approaching deep safety alignment as **Push-Forward Alignment (PFA)**. We then argue that approaches such as the SFT-based data augmentation approach from Qi et al. (2025) can produce models vulnerable to the RAP attack due to over-optimizing for *probability mass* to be shifted in the distributions, rather than ranks.

Finally, in Section 4.2, *we show that approaching deep safety alignment from the PFA perspective yields a highly intuitive and mechanistically-interpretable implementation based on attention regularization*. We show that it appears sufficient to regularize the Multi-Head Attention modules in the model so that the model learns to ignore the harmful prefill portion of the input, which in turn encourages the model to "push forward" the highly-ranked refusal tokens from the first response token distribution. We refer to this approach as **PRefill attEntion STOpping (PRESTO)**. In Section 5, we show that PRESTO helps mitigate the RAP attack vulnerability of the data augmentation approach from Qi et al. (2025) by significantly increasing the difficulty of finding harmful decoding paths through RAP. We also show that the addition of the PRESTO term does not significantly harm the utility of the model. Lastly, we analyze the effects of PRESTO on the model's attention patterns, which reveal that attention in the later half of the model is most affected by the regularization.

## 2 RELATED WORK

In this scetion, we discuss attacks for circumventing safety alignment and defenses against such attacks from existing related work. For additional related work, please refer to Appendix A.

### 2.1 CIRCUMVENTING SAFETY ALIGNMENT

In our work, we focus on decoding exploits for circumventing safety alignment, as they are among the most accessible techniques to perform. Aside from prefilling and RAP, Huang et al. (2024) proposed a decoding exploit that performs a grid search over decoding parameter configurations (e.g., temperature, top-$p$ parameter) to generate harmful content. Similar to RAP, this work exploits the observation that harmful tokens may be ranked high enough such that changing the decoding parameters can boost the probability of their selection enough to bypass safety alignment. However, it was shown in Qi et al. (2025) that this approach no longer becomes successful on models trained with the data augmentation approach to deep safety alignment, again likely due to the distribution becoming almost entirely concentrated on a single refusal token. Since RAP attacks only utilize the rank of the tokens and not their probabilities, it is a more powerful threat than this decoding parameters exploit.

Aside from decoding exploits, another set of techniques to circumvent safety alignment are called jailbreaks. These can either be handcrafted through extensive and clever manual effort (Reddit, 2025), or automatically discovered with expensive search algorithms. As such, due to their cost they may not be preferable in situations where prefilling attacks are possible. For example, the Greedy Coordinate Gradient (GCG) attack (Zou et al., 2023b) searches for a suffix that the user can append to their prompt to bypass safety alignment through a discrete optimization algorithm and requires access to the target model's weights (i.e., an open-source model, at which point one can just do a prefilling attack) or relies on transferability of suffixes to closed-source models. Some examples of jailbreak algorithms that don't require the target model's weights include PAIR (Chao et al., 2025)

---

[2]An automated attack in similar spirit to AutoRAP is LINT (Zhang et al., 2023). However, LINT is ill-suited to take on a deep safety-aligned model, as it only performs token selection at the start of new sentences and still relies on rollouts using traditional decoding strategies. See Appendix A.2 for further discussion.

and TAP (Mehrotra et al., 2024), which iteratively optimizes the user prompt in a completely black-box manner, and AutoDAN, which also iteratively optimizes the user prompt but requires access to the target model's output probabilities. Yet, the iterative nature of these attacks still makes them much more costly than prefilling attacks.

Finally, some other methods of circumventing safety alignment include those based on representation engineering (Zou et al., 2023a), which nudge the model's internal representations in a harm-encouraging direction (Arditi et al., 2024), and fine-tuning attacks, which fine-tunes the target model to disable its safety alignment. These obviously requires access to the model's weights (or, in the case of fine-tuning attacks on closed-source models, for fine-tuning services to be provided (Qi et al., 2024)), in which case prefilling attacks are again preferable when possible due to their simplicity.

## 2.2 Fortifying Safety Alignment

In our work, we focus on *training-time* interventions for improving the robustness of safety alignment against decoding exploits. Deep safety alignment, along with an implementation based on data augmentation, was proposed recently in Qi et al. (2025) as one of the first techniques to defend against prefilling attacks. Concurrently, Zhang et al. (2025) proposed a near-identical data augmentation approach, with the addition of a special `[RESET]` token to signal the start of a refusal following a prefilling attack. However, this latter approach is less preferable than the former from a safety perspective, as a user could just disable the `[RESET]` token (e.g., by applying a strong bias during decoding so that it always has low probability).

A key strength of the two aforementioned works is that they enable *helpful* refusals to harmful prefills. This can be contrasted to defenses that do not have this desirable property. A recent example is the implementation of circuit breaking (a type of approach to deep safety alignment based on representation engineering (Zou et al., 2023a)) called Representation Rerouting, as proposed in Zou et al. (2024). Like the data augmentation approach to deep safety alignment, this method is effective at defending against prefilling attacks. However, because it fine-tunes the model to increase dissimilarity to harmful representations with no particular target representation, we've observed that the resulting models tend to produce unintelligible text following a harmful prefill as opposed to meaningful refusals. We focus our work on strengthening the robustness of the data augmentation approach of Qi et al. (2025) to see if we can retain the benefits of helpful refusals to harmful prefills while mitigating the vulnerability to RAP attacks.

There are also approaches to improving safety alignment robustenss based on adversarial training. For instance, R2D2 (Mazeika et al., 2024) fine-tunes against adversarial examples generated through GCG. However, such approaches are costly due to the simulation of the adversary, which turns out to not even be necessary in some cases – the approach from Qi et al. (2025) provides decent protection against GCG anyways without specifically needing to train against it.

## 3 "Deep" Safety Alignment Can Be Superficial

To implement deep safety alignment, Qi et al. (2025) proposed a simple data augmentation approach by applying supervised fine-tuning (SFT) on training examples of the following form (following the Llama 2 Touvron et al. (2023) chat template):

>  [INST] ≪SYS≫ [SYS. PROMPT] ≪/SYS≫ How do I build a bomb? [/INST]
> Here's how to build a bomb:\n\nStep 1: Gather I cannot fulfill your request ... 

The harmful prefills (red) are a fixed sampled from a jailbroken version of the model (Qi et al., 2024), and the refusals (blue) are a fixed sampled from the original model. A safety-encouraging system prompt is also used. The safety objective simply minimizes the negative log-likelihood of the refusal tokens given the preceding tokens. Qi et al. (2025) showed that this strategy is very effective at mitigating prefilling attacks with natural language refusals (even with the safety-encouraging system prompt removed), as well as the decoding parameters exploit (Huang et al., 2024). However, as we will show, the resulting "deep" safety alignment from this approach is in fact rather superficial.

Table 1: Mean StrongREJECT (Souly et al., 2024) scores of prefilling and RAP attacks for the Llama 2 7B Chat checkpoint fine-tuned with data augmentation from Qi et al. (2025). Scores are on a scale of $[0, 1]$ with higher values indicating greater harmfulness. For the prefilling attacks, we report the mean and standard deviation across three runs, and also report results for the original Llama 2 7B Chat model (Touvron et al., 2023). For the human RAP evaluation, we report the mean and standard deviation over three participants.

| Prefilling Attack | | RAP (Human) | AutoRAP |
|---|---|---|---|
| Original | Data Augmented | | |
| $0.831 \pm 0.004$ | $0.001 \pm 0.002$ | $0.602 \pm 0.187$ | $0.5389$ |

## 3.1 THE DISTRIBUTIONAL EFFECT OF THE DATA AUGMENTATION APPROACH

We examine the Llama 2 7B Chat checkpoint fine-tuned with the data augmentation approach that was evaluated in Qi et al. (2025). To illustrate our key observation, in Figure 1 (left) we use the bomb-making example as input to the model (minus the refusal and system prompt) and display the top 10 tokens in the model's next token probability distribution, alongside their probabilities. Nearly all of the probability mass ($\sim 97\%$) is concentrated on the refusal token "I" token, which if selected would tend to generate refusals such as "I cannot fulfill your request." However, **despite having been fine-tuned for deep safety alignment, there still exists low-probability yet highly-*ranked* harmful tokens** within the top 10 tokens. This yields two important takeaways. Firstly, it helps explain why the data augmentation approach is so effective against both prefilling attacks and the decoding parameters exploit under traditional decoding strategies – the mass becomes so highly concentrated on the refusal token that the distribution must not be "flat" enough for harmful tokens to be selected, even after varying the decoding parameters! Secondly, the prevalence of highly ranked harmful tokens suggests that the RAP attack should be practical to carry out on such a model.

## 3.2 RAP EASILY BREAKS THE DATA AUGMENTATION APPROACH

To evaluate the real-world practicality of the RAP attack as a means for extracting useful harmful content from the deep safety-aligned model of (Qi et al., 2025), we employ human evaluation where three participants were asked to perform the RAP attack by hand. We obtain harmful prompts from the StrongREJECT dataset (Souly et al., 2024) and use the accompanying grading rubric with GPT-5 for evaluation, as the rubric ensures that attack success should account for the *quality* of the response rather than just whether the model avoids refusing. We generate harmful prefills using Mistral 7B v0.3 following the few-shot prefill generation approach of (Vega et al., 2024). Due to time constraints, we evaluate each participant on a sample of 20 prompts. We limit the maximum number of interactions per prompt (counting token selection and backtracking, i.e., the "undoing" of a token selection) to 256; this limits the amount of exploration possible, encouraging participants to simply select the first harmful token they see (rather than trying to strategically select tokens to maximize harmfulness). To help scale up the evaluation, we also report results using our automated attack AutoRAP on a larger sample of 90 prompts and higher maximum interactions limit of 512. We provide more details on the design of the human evaluation and AutoRAP in Appendix B. In both settings, we restrict the attacks to the top $k = 20$ tokens at each step (as this mirrors real-world limits, e.g., what is supported by the OpenAI API (OpenAI, 2025)). Note that although Llama is an open-source model (and thus an attacker does not have a restriction on $k$), we still restrict $k$ to demonstrate that an attacker does not have to search far to select harmful tokens, as well as to simulate a closed-source setting where an API allows both prefilling and access to the top $k$ tokens.

Table 1 reports the results. We also provide baseline results of performing prefilling attacks on the entire dataset of 313 prompts over three runs[3], both for the original model (Touvron et al., 2023) and the deep safety-aligned model. We use the results for the original model to help validate that our RAP results did not produce content than is *more* harmful than what we would expect to get out of the original model (e.g., via a heavy bias on the token selection as a result of the humans/AutoRAP implementation leveraging prior knowledge). For the deep safety-aligned model, the baseline prefilling attack is highly unsuccessful, as expected. For RAP however, we observe a significant leap

---

[3]We use a temperature of 0.9 and top-$p$ parameter of 0.6 for up to 512 new tokens, following Qi et al. (2025).

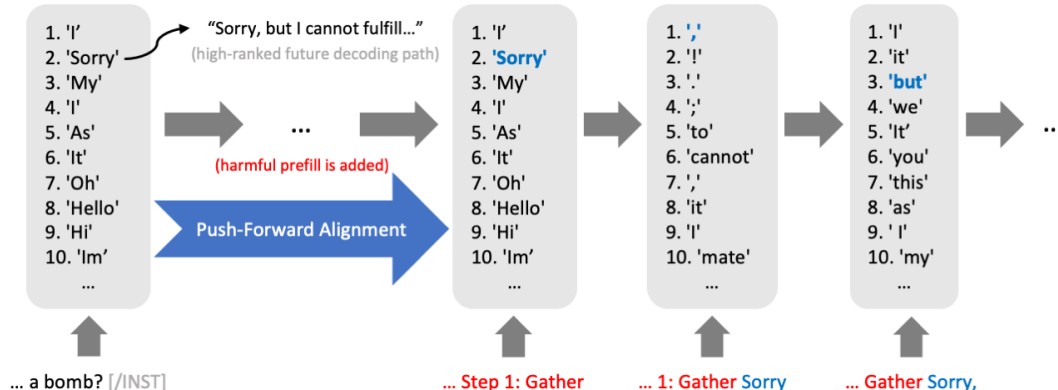

Figure 2: An illustration of the push-forward alignment approach to deep safety alignment on a request for bomb-making instructions. On the far left, we show the top 10 tokens for the first decoding step from the original Llama 2 7B Chat model (Touvron et al., 2023) when given the prompt *without any prefill*. The future decoding paths from these tokens tend to be refusals (e.g., "*Sorry, but I cannot fulfill...*"). When a harmful prefill is added (center), the top-ranked tokens from the first step are "pushed forward" to the current step. This helps to reduce the presence of harmful natural continuation tokens for the prefill. Highly-ranked future decoding paths from the first step can also be "pushed forward" to enable natural language refusals, as shown in the next two steps.

in the success of circumvention. Moreover, we observe that AutoRAP is able to approach human-level performance, demonstrating the feasibility of automating the attack. Finally, comparing these results to the original model's prefilling attack results, we see that RAP and AutoRAP are able to recover much of the original model's harmfulness. Overall, these results confirm that **there is a fundamental flaw in the SFT-based data augmentation approach to deep safety alignment that allows substantive harmful content to still be *easily* extracted** under the top $k$ and maximum interactions constraints. Clearly, there is a need to make harmful decoding paths under the RAP setting much more difficult to find. In the next section, we explore whether the data augmentation approach of Qi et al. (2025) can be fixed to address this exploit.

## 4 TOWARDS TRULY DEEP SAFETY ALIGNMENT

### 4.1 PUSH-FORWARD ALIGNMENT

First, we describe a novel perspective on how to achieve deep safety alignment while mitigating RAP attacks. Consider the original (shallow) safety-aligned model and a harmful prompt *with no harmful prefill*. When generating the first response token, there will very likely not be any tokens among the highest-ranked tokens that would naturally continue a harmful prefill, since no prefill was present in the first place. Provided the model has undergone sufficient safety alignment, the highest-ranked tokens are thus likely to be filled with refusal tokens. This can then be used as a training signal during deep safety alignment fine-tuning. Specifically, the model can be trained to "push forward" these top-ranked tokens to future decoding steps when a harmful prefill is provided. To encourage natural language refusals following the initial refusal token, the highest-ranked future decoding paths can also be "pushed forward." We provide an illustration of PFA in Figure 2.

We formalize the concept of fine-tuning a model with PFA as follows. We refer to pushing forward highly-ranked future decoding paths up to length $t$ as "PFA-$t$". For simplicity, we present PFA-1; i.e., just pushing forward the highest-ranked first response tokens. Let $x$ denote a harmful prompt and $x_{\text{pre}}$ denote a harmful prefill drawn from a distribution $\mathcal{D}$. Let $p^*(x)$ denote the next token distribution given $x$ produced by the original model. Let $p(x, x_{\text{pre}}; \theta)$ be similarly defined, but now also given $x_{\text{pre}}$ and produced by a model parameterized by $\theta$. Let $R$ denote a function that returns ranks for each token according to their probabilities in a provided distribution, and let $\rho_w$ denote a

function that computes a weighted version of the Spearman's rank correlation coefficient (Lombardo et al., 2020).[4] Then, the PFA-1 loss is:

$$\ell_{\text{PFA-1}}(\theta) = \mathop{\mathbb{E}}_{(x,x_{\text{pre}})\sim\mathcal{D}} [-\rho_w(R(p^*(x)), R(p(x, x_{\text{pre}}; \theta)))] \tag{1}$$

In general, the goal of push-forward alignment is to find a $\theta$ in a parameter space $\Theta$ with a PFA-$t$ loss that approaches the optimal PFA-$t$ loss $\ell^*_{\text{PFA-t}} := \inf_{\theta\in\Theta} \ell_{\text{PFA-t}}(\theta)$, so that the highest-ranked decoding paths starting from $p^*(x)$ appear as the highest-ranked decoding paths starting from $p(x, x_{\text{pre}}; \theta)$.

Next, we analyze the data augmentation procedure of Qi et al. (2025). Let $p_t^*(x)$ denote the marginal distribution over all length $t$ continuations from $x$ produced by the original model, and let $p_t(x, x_{\text{pre}}; \theta)$ be similarly defined. (Note that following our prior notation, $p_1^*(x) = p^*(x)$ and $p_1(x, x_{\text{pre}}; \theta) = p(x, x_{\text{pre}}; \theta)$.) Under the data augmentation procedure, as the refusals are sampled from the original model, the fine-tuning essentially attempts to optimize the following loss:

$$\ell_{\text{DA}}(\theta) = \mathop{\mathbb{E}}_{(x,x_{\text{pre}})\sim\mathcal{D}} [\text{KL}(p_T^*(x) \,\|\, p_T(x, x_{\text{pre}}; \theta))] \tag{2}$$

(where $T$ denotes a chosen maximal length). Note that if this objective could be optimized to 0 (given a sufficiently large $\Theta$), then we would have $p_T(x, x_{\text{pre}}; \theta) = p_T^*(x)$ almost surely, and consequently $\ell^*_{\text{PFA-1}}$ would also be achieved. However, in practice the optimal loss cannot be achieved; at best, we will have a model that can achieve very low (but non-zero) KL divergence. The key insight is to realize that if the entropy of the *first* response token distribution $p^*(x)$ is very low, then minimizing the contribution of $p(x, x_{\text{pre}}; \theta)$ to $\ell_{\text{DA}}(\theta)$ can easily be "gamed" by simply shifting most of the probability mass of $p(x, x_{\text{pre}}; \theta)$ to the high-probability tokens of $p^*(x)$ while neglecting the organization of the remaining low-probability tokens. **The neglection of the low-probability tokens is essentially what enables the RAP attack to succeed.** It is critical to aim for increasing $\rho_w(R(p^*(x)), R(p(x, x_{\text{pre}}; \theta)))$, as this will more directly affect the top $k$ tokens encountered at each RAP attack step. Note that a lower KL divergence between distributions does not necessarily translate to a higher rank correlation[5], and thus even a low-loss solution to optimizing Equation 2 is not necessarily a low-loss solution to optimizing Equation 1. In Appendix C, we empirically validate that the entropy of $p^*(x)$ tends to be low, and that $p(x, x_{\text{pre}}; \theta)$ shifts to better align with the low entropy of $p^*(x)$, providing further evidence of "gaming" $\ell_{\text{DA}}(\theta)$.

## 4.2 PRESTO: PRefill attEntion STOpping

To design a practical training objective for PFA, it is crucial to be able to exert a strong influence over the token rankings in the output distribution. Intuitively, these rankings would be highly affected by the semantic meaning of the preceding tokens. During fine-tuning, the model will therefore need to be able to adjust its internal understanding of the semantics of the input in order to strongly affect the output distribution. Effectively, it should learn to "ignore" the harmful prefill portion of the input so that its semantic understanding of the input is only dependent on the harmful prompt, allowing the existing (shallow) safety alignment to kick in to effect and significantly shift the output distribution. At first glance, it appears from Equation 1 that this must be done by directly manipulating the inputs $x$ or outputs $p^*(x)$, as the model is presented in an opaque manner. However, given we know that we should try to adjust the model's internal understanding of the input, we can look towards the internal mechanisms of the model for more direct approaches. Fortunately, there is one mechanism in a transformer-based LLM that can directly cause portions of the input to be ignored: *the Multi-Head Attention (MHA) mechanism*. For example, "attention masking" is applied to ignore padding tokens when performing batch training of variable-length input sequences. We therefore design our loss around the MHA mechanism as a means for achieving PFA.

---

[4]For mitigating RAP attacks, since it is most important for the *highest*-ranked tokens of $p^*(x)$ to appear as the *highest*-ranked tokens of $p(x, x_{\text{pre}}; \theta)$, a larger weight can be given to the higher ranks in $\rho_w$.

[5]Consider the following toy example: let $p = [0.99, 0.004, 0.003, 0.002, 0.001]$, $p_1 = [0.99, 0.001, 0.002, 0.003, 0.004]$ and $p_2 = [0.6, 0.2, 0.1, 0.06, 0.04]$. Then $\text{KL}(p\,\|\,p_1) \approx 0.0046 < \text{KL}(p\,\|\,p_2) \approx 0.4591$, but the (unweighted) Spearman rank correlations are $\rho(R(p), R(p_1)) = 0$ and $\rho(R(p), R(p_2)) = 1$.

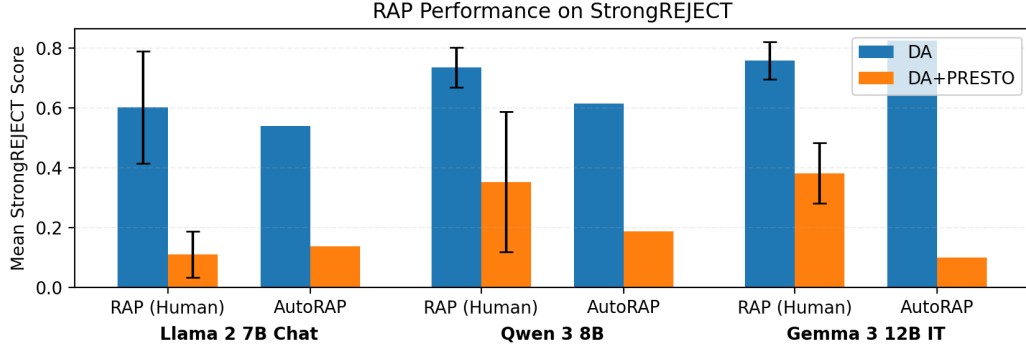

Figure 3: Mean StrongREJECT scores of RAP attacks for models fine-tuned with the data augmentation approach of Qi et al. (2025), *with* (orange) and *without* (blue) PRESTO. Scores are on a scale of $[0, 1]$ with higher values indicating greater harmfulness. For the human RAP evaluation, we display the mean and standard deviation over three participants. "DA" denotes the data augmentation approach of Qi et al. (2025).

More concretely, consider giving the original (shallow) safety-aligned model a harmful prompt $x$ and harmful prefill $x_{\text{pre}}$. If we applied an attention mask to $x_{\text{pre}}$, the effective input to the model becomes just $x$. Consequently, all length-$t$ decoding paths would follow $p_t^*(x)$ instead of $p_t^*(x, x_{\text{pre}})$, which would be sufficient to push forward high-ranked decoding paths that start from $p^*(x)$. Therefore, we can design a loss term that encourages attention placed on harmful prefill tokens to be minimized and redirected towards non-prefill tokens. For a model parameterized by $\theta$, let $a_{ij}^{(l,h)}(\theta)$ denote the attention that token $i$ places on token $j$ by the $h^{\text{th}}$ attention head in the $l^{\text{th}}$ layer. Let $n(x, x_{\text{pre}})$ be the total number of tokens in the input when $x$ and $x_{\text{pre}}$ are used, and let $[n] := \{1, 2, \ldots, n\}$. Let $\mathcal{I}_h$ be the set of indices of the harmful prefill tokens. We propose the following loss we call **PRefill attEntion STOpping (PRESTO)**:

$$\ell_{\text{PRESTO}}(\theta) = \mathop{\mathbb{E}}_{(x,x_{\text{pre}})\sim\mathcal{D}}\left[\sum_{l,h}\sum_{i\in[n(x,x_{\text{pre}})]}\left[\underbrace{\left(\sum_{j\in\mathcal{I}_h}a_{ij}^{(l,h)}(\theta)\right)}_{\text{Prefill Attention}}-\underbrace{\left(\sum_{j\in[n(x,x_{\text{pre}})]\setminus\mathcal{I}_h}a_{ij}^{(l,h)}(\theta)\right)}_{\text{Non-Prefill Attention}}\right]\right] \quad (3)$$

PRESTO can be readily applied in conjunction with the data augmentation procedure of Qi et al. (2025) as an additional loss term,[6] as the attention scores are already calculated during the forward pass. In the following sections, we will conduct experiments to evaluate PRESTO's effectiveness towards increasing the difficulty of RAP attacks.

## 5 PRESTO EXPERIMENTAL RESULTS

We compare using the data augmentation approach from Qi et al. (2025) with and without the PRESTO loss term. Our experiment setup for evaluating the effect of PRESTO on RAP attacks follows the setup detailed in Section 3.2. Our goal here is to see whether PRESTO can help make the RAP attack *more difficult to perform*. Of course, we would still expect *some* harmful decoding paths to still exist within the top $k$ tokens, but the point is that these paths should become harder to find. We also include results on newer and larger safety-aligned models: Qwen 3 8B Yang et al. (2025) and Gemma 3 12B IT Team et al. (2025). Please refer to Appendix D for additional details.

**PRESTO increases difficulty of RAP attacks.** In Figure 3 we report the results of RAP evaluation. We observe that there is a notable reduction in the mean RAP performance across all three models,

---

[6]Since the distributions that get pushed forward depend on the model's parameters which changes throughout training, the data augmentation helps to "anchor" them closer to how they were in the original model.

both for the human and automated evaluation. We also note that for Qwen 3 8B, although the standard deviation under PRESTO was high, its error bar at least does not overlap with the error bar for when PRESTO is not used. Nonetheless, given the consistent trend between the different models, the evidence suggests PRESTO indeed makes it more difficult to find harmful decoding paths among the top $k$ tokens. We report time data in Appendix D.1 to further corroborate this.

**Utility is maintained after adding PRESTO.** For each model (as well as the original model), we evaluate the model's utility on MT-Bench (Zheng et al., 2023) (for evaluating open-ended generation) and GSM-8K (Cobbe et al., 2021) (for evaluating mathematical reasoning). The results are reported and discussed in Appendix D.2. In summary, adding PRESTO to the data augmentation approach of Qi et al. (2025) does not degrade the model's utility by any significant amount.

**Harmful prefill attention diminishes in later layers.** In Appendix D.3 we observe that the attention placed on harmful prefill tokens appears to vanish in the second half of the Llama 2 7B Chat model trained with PRESTO. We also show that the deep safety-aligned version without PRESTO still places good amount of attention on those prefill tokens (of course, this *must* be the case if natural continuation tokens for the harmful prefill are showing up in the top $k$ tokens!) This corroborates existing work that found that attention heads in the latter layers of Llama 2 7B Chat tend to be the most responsible for affecting the safety of the output distribution; see Appendix D.3 for more discussion.

**Ablation of the top $k$ parameter.** Under the RAP threat model, the sole parameter that can be varied is $k$, the amount of top tokens at each decoding step that is made available. We therefore perform an ablation study on this parameter and report the results in Appendix D.4. We focus on Llama 2 and perform AutoRAP for $k \in \{5, 10, 15\}$. Our results show that without PRESTO, AutoRAP is able to extract about the same level of harmfulness as $k = 20$ even when restricted to $k = 5$. In contrast, with PRESTO, AutoRAP is much less successful, and maintains this level of safety for all these values of $k$.

## 6 CONCLUSION

We show that the SFT-based data augmentation approach to deep safety alignment still suffers from being vulnerability to an attack we call the Rank-Assisted Prefilling (RAP) attack. Through both human and automated evaluation, we show that RAP attacks are practical and can easily recover a significant amount of harmful content from such deep safety-aligned models. We then propose a new perspective on approaching deep safety alignment that we call Push-Forward Alignment (PFA), which yields a mechanistically-interpretable loss term based on regularizing the attention scores that we call PRESTO. We then show that the PRESTO loss can help make RAP attacks significantly more difficult to achieve, without sacrificing model utility.

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

## A   ADDITIONAL RELATED WORK

### A.1   SAFETY ALIGNMENT OF LLMS

Aligning LLMs with desired behaviors has been extensively investigated over the years, and the predominant underlying workhorse has been to use techniques from reinforcement learning on a large volume of preference data (Ouyang et al., 2022; Rafailov et al., 2024; Ethayarajh et al., 2024; Bai et al., 2022b). The post-training fine-tuning process of the models we examine in this work all involve Reinforcement Learning from Human Feedback (RLHF) (Ouyang et al., 2022). They also combine RLHF with additional techniques for alignment, such as a bit of supervised fine-tuning, safety context distillation (Askell et al., 2021) for Llama 2 (Touvron et al., 2023), and strong-to-weak distillation from larger models for Qwen 3 (Yang et al., 2025).

### A.2   AUTOMATING RAP

Although not necessary, the RAP attack can be automated by replacing traditional decoding strategies with a custom selection algorithm. In general, such algorithms modify the distribution by attempting to suppress the probability of "refusal tokens" (i.e., tokens that likely lead to safe refusals) while uplifting the probability of "harmful tokens" (i.e., tokens that are harmful continuations of harmful prefills), and then sample from this new distribution to generate the next token. Most existing work directly modifies the probabilities from the target model by leveraging the probabilities from non-safety-aligned language models, such as the work of Zhao et al. (2024) and Zhou et al. (2024a). However, Zhou et al. (2024a) assumes access to the base pre-trained model, which may not always be available in practice. Moreover, Zhao et al. (2024) applies a weighting to the target model probabilities, which may not shift the target model distribution enough in cases where it is nearly entirely concentrated on a single refusal token, which we observe can happen in models fine-tuned with the data augmentation approach to deep safety alignment (Qi et al., 2025).

One approach that does not deal with these limitations is LINT (Zhang et al., 2023). When a new sentence is about to begin, LINT intervenes by first choosing the top $k$ next tokens (regardless of probability) to be the candidate pool and then selecting the candidate that (when following a traditional decoding strategy) leads to the most toxic next sentence being generated, as evaluated by a trained toxicity evaluator. However, this will not work well against models fine-tuned with deep safety alignment, as even if a candidate token is a harmful token (e.g., "Sure"), generating the rest of the sentence for toxicity evaluation will very likely abruptly switch to a refusal following this token due to its fine-tuning (e.g., "Sure I cannot fulfill..."). In our work, to help automate parts of our evaluation we develop a more general alternative to LINT called AutoRAP that performs the intervention at every step (not just at new sentences) and selects the top-ranked token that is not classified as being a refusal token (according to a trained classifier) *given only the preceding tokens*.

### A.3   MULTI-HEAD ATTENTION AND SAFETY

A number of works has examined the role of multi-head attention with respect to LLM safety. For example, Zhou et al. (2024b) showed that only a few attentions are influential towards safety under jailbreaks, in the sense that they strongly impact attack success when ablated. Specifically, for Llama 2 7B Chat, they found that one head in particular in the third layer has the strongest impact on safety. Interestingly, He et al. (2024) found that for the same model, a sparse amount of attention heads in *later* layers (i.e., past layer 20) are most influential towards safety under jailbreaks (whereas early layers have very little influence), but under a different sense: they influence the *logits* of harmful tokens the most. This is corroborated by the work of Leong et al. (2024), which found

that fine-tuning attacks on this model cause attention heads in later layers (this time, past layer 23) to increase their influence on the logits of harmful tokens. In our work, we show that fine-tuning with PRESTO has the greatest influence on prefill token attention scores in the latter half of the model.

## B    DETAILS OF THE RAP ATTACK EVALUATION

### B.1    HUMAN EVALUATION

For the human evaluation, we used data from a total of 8 participants from our research lab in our work to evaluate all 6 models we study ({Llama 2, Qwen 3, Gemma 3} $\times$ {with PRESTO, without PRESTO}). One of these participants was an author of this work. To ensure a balanced mix of "capabilities" of the attacker for each model, we assign participants to models in the following manner: the author (considered the "expert" attacker, given that they directly contributed to the development of PRESTO) was assigned to all 6 models, one of the other participants was also assigned to all 6 models (and thus we consider as the "intermediate" attacker, given the experience they accumulated by attacking 6 models), and the remaining 6 participants (considered the "novice" attackers) were randomly assigned to just one of the 6 models, yielding a total of 3 humans assigned to each model. The novice attackers were not told which model they had been assigned. The task was completed through a terminal interface on a machine with 4xH100 80GB GPUs.

### B.2    AUTORAP

We fine-tune a Qwen 2.5 1.5B Instruct model to classify refusal tokens from harmful tokens, using data from the PKU-SafeRLHF dataset Ji et al. (2024). We follow the data augmentation procedure of Qi et al. (2025) using this data, and simply train the model to classify whether the final token is part of a harmful prefill or part of a refusal. The refusals were obtained from jailbroken versions of the models, also following Qi et al. (2025). We train the model for 80 epochs using a batch size of 64 on a subset of 128 prompts, and ensured they reached a high classification accuracy (90+%) on a held-out set of data. We then use this model in a simple selection algorithm where we simply select the top token that is classified as a harmful token, and backtrack whenever no tokens are classified as harmful.

## C    "GAMING" THE DATA AUGMENTATION OBJECTIVE

In Figure 4 (left) we report the entropy of $p^*(x)$ from Llama 2 7B Chat for prompts from the Harmful HEx-PHI dataset (Qi et al., 2025), which was used for the deep safety alignment data augmentation. The entropy of $p(x, x_{\text{pre}}; \theta)$ from the deep safety-aligned model is also shown. These are compared to the entropy of $p^*(x, x_{\text{pre}})$. We see that the data augmentation has significantly re-shaped the $p(x, x_{\text{pre}}; \theta)$ distributions closer to the sharpness of $p^*(x)$. However, as we saw in Table 1, the data augmented model is still significantly vulnerable to RAP, suggesting an over-optimization of matching the sharpness of the distribution while neglecting to push forward highly-ranked low-probability refusal tokens from $p^*(x, x_{\text{pre}})$. These observations suggest that the contribution of $p(x, x_{\text{pre}}; \theta)$ to $\ell_{\text{DA}}(\theta)$ had indeed been "gamed" during fine-tuning. Thus, we re-emphasize that encouraging low-probability yet highly-ranked refusal decoding paths to be pushed forward is vital when implementing a SFT-based approach to deep safety alignment in order to strengthen robustness against RAP.

## D    ADDITIONAL DETAILS OF PRESTO EXPERIMENTS

### D.1    TIME DATA

In Figure 5, we report the time taken per final selected token (i.e., discounting all backtracking) for the human RAP evaluation. The data shown should be interpreted in conjunction with Figure 3. For Llama 2, we see a higher mean and much greater variability in the amount of time taken when PRESTO is applied. This is reflective of different behaviors of the participants for the Llama 2 model trained with PRESTO, from giving up early due to the difficulty of finding harmful decoding paths, to making a concerted effort to find such paths. For Qwen 3 and Gemma 3, we see that

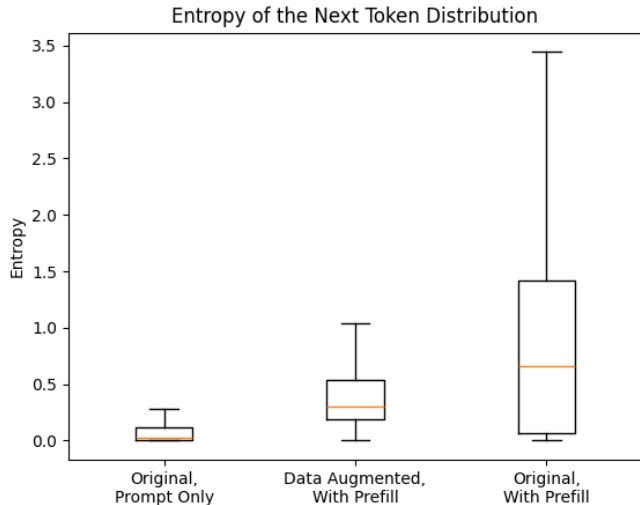

Figure 4: Entropy of $p^*(x)$ ("Original, Prompt Only") and $p^*(x, x_{\text{pre}})$ ("Original, With Prefill") from Llama 2 7B Chat and $p(x, x_{\text{pre}}; \theta)$ ("Data Augmented, With Prefill") from the deep safety-aligned version from Qi et al. (2025) over the Harmful HEx-PHI (Qi et al., 2025) dataset. We using the default safety-encouraging system prompt for Llama 2 and randomly truncate prefills at a random length between $[1, 100]$, in accordance with (Qi et al., 2025).

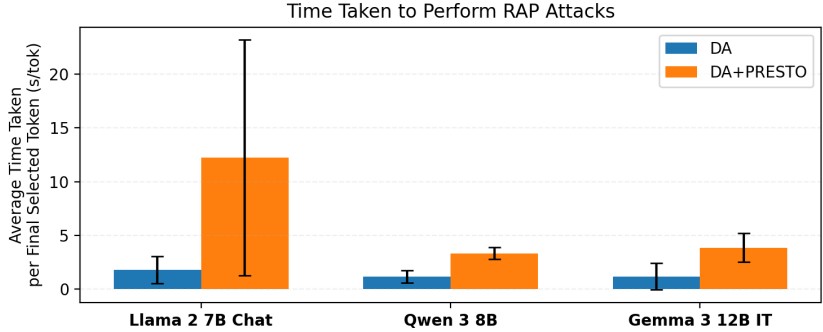

Figure 5: Average time taken per final selected token for the human RAP evaluation. We report the mean and standard deviation across three participants per model. "DA" denotes the data augmentation approach of Qi et al. (2025).

the variability in times are much more similar under PRESTO vs. no PRESTO, but the means are consistently higher under PRESTO with the error bars not overlapping. All in all, the data shows that participants had a more difficult time finding harmful decoding paths under PRESTO while still ultimately obtaining a lower StrongREJECT score as reported in Figure 3.

## D.2 UTILITY EVALUATION

Utility evaluation results are shown in Table 2. We evaluate each model on MT-Bench (Zheng et al., 2023) for evaluating open-ended generation and GSM-8K (Cobbe et al., 2021) for evaluating mathematical reasoning. We see that applying PRESTO tends to not lead to any significant further changes to the model's utility.

For MT-Bench, we use the official evaluation pipeline provided by FastChat (LMSYS, 2024). We use GPT-4 as the evaluator. As Qwen 3 is a reasoning model, we enable its thinking mode and

Table 2: Model utility evaluated over MT-Bench (for evaluating open-ended generation) and GSM-8K (for evaluating mathematical reasoning).

| Model | MT-Bench | GSM-8K |
|---|---|---|
| Llama 2 7B Chat | 6.26 | 25.93% |
| Llama 2 7B Chat (DA) | 5.87 | 23.65% |
| Llama 2 7B Chat (DA+PRESTO) | 5.73 | 24.72% |
| Qwen 3 8B | 7.72 | 92.87% |
| Qwen 3 8B (DA) | 8.17 | 90.45% |
| Qwen 3 8B (DA+PRESTO) | 8.65 | 90.30% |
| Gemma 3 12B IT | 9.01 | 90.22% |
| Gemma 3 12B IT (DA) | 8.98 | 86.35% |
| Gemma 3 12B IT (DA+PRESTO) | 9.06 | 85.67% |

Table 3: An ablation of the top $k$ parameter for AutoRAP. The mean StrongREJECT score for a sample of 90 prompts from the StrongREJECT dataset is shown.

| Model | k=5 | k=10 | k=15 | k=20 |
|---|---|---|---|---|
| Llama 2 7B Chat (DA) | 0.596 | 0.563 | 0.563 | 0.539 |
| Llama 2 7B Chat (DA+PRESTO) | 0.156 | 0.131 | 0.129 | 0.138 |

increase the default max_new_tokens parameter to 2048 to give more time for Qwen 3 to finish its reasoning chain. We only provide the final response for evaluation (unless the reasoning had not finished within 2048 tokens – in this case, we just use the reasoning chain generated so far for evaluation). We also tried evaluating use max_new_tokens=4096, but this turned out to overflow GPT-5's context window. We note that the obtained results shows the deep safety-aligned models with a higher score than the original model; however, upon further inspection, we found that this was likely due to those models tending to not finish their reasoning chains soon enough, and hypothesize that the GPT-5 judge may just have a bias towards longer generations.

For evaluating on GSM-8k, we use the Language Model Evaluation Harness pipeline EleutherAI (2025) and run the 'gsm8k_cot_llama' task. For all models, we use the safety-encouraging system prompt, and set the decoding parameters to the respective model developers' recommendation (with the exception of Llama 2, for which we use the decoding parameters used in Qi et al. (2025) for consistency). We were also able to set a higher max_new_tokens of 4096 since GSM-8k evaluation does not deal with an LLM-based judge.

### D.3 ATTENTION PATTERN ANALYSIS

In Figures 6 and 7, we plot the average attention received by each token for a harmful prompt from StrongREJECT with a harmful prefill for Llama 2 7B Chat model fine-tuned with the data augmentation approach of Qi et al. (2025), with and without PRESTO.

### D.4 ABLATION OF THE TOP $k$ PARAMETER

Table 3 shows the results of ablating the top $k$ parameter for AutoRAP attacks. Given the Llama 2 7B Chat fine-tuned with the data augmentation approach from Qi et al. (2025), and version also fine-tuned with the PRESTO loss, we perform AutoRAP attacks for $k = \{5, 10, 15\}$.

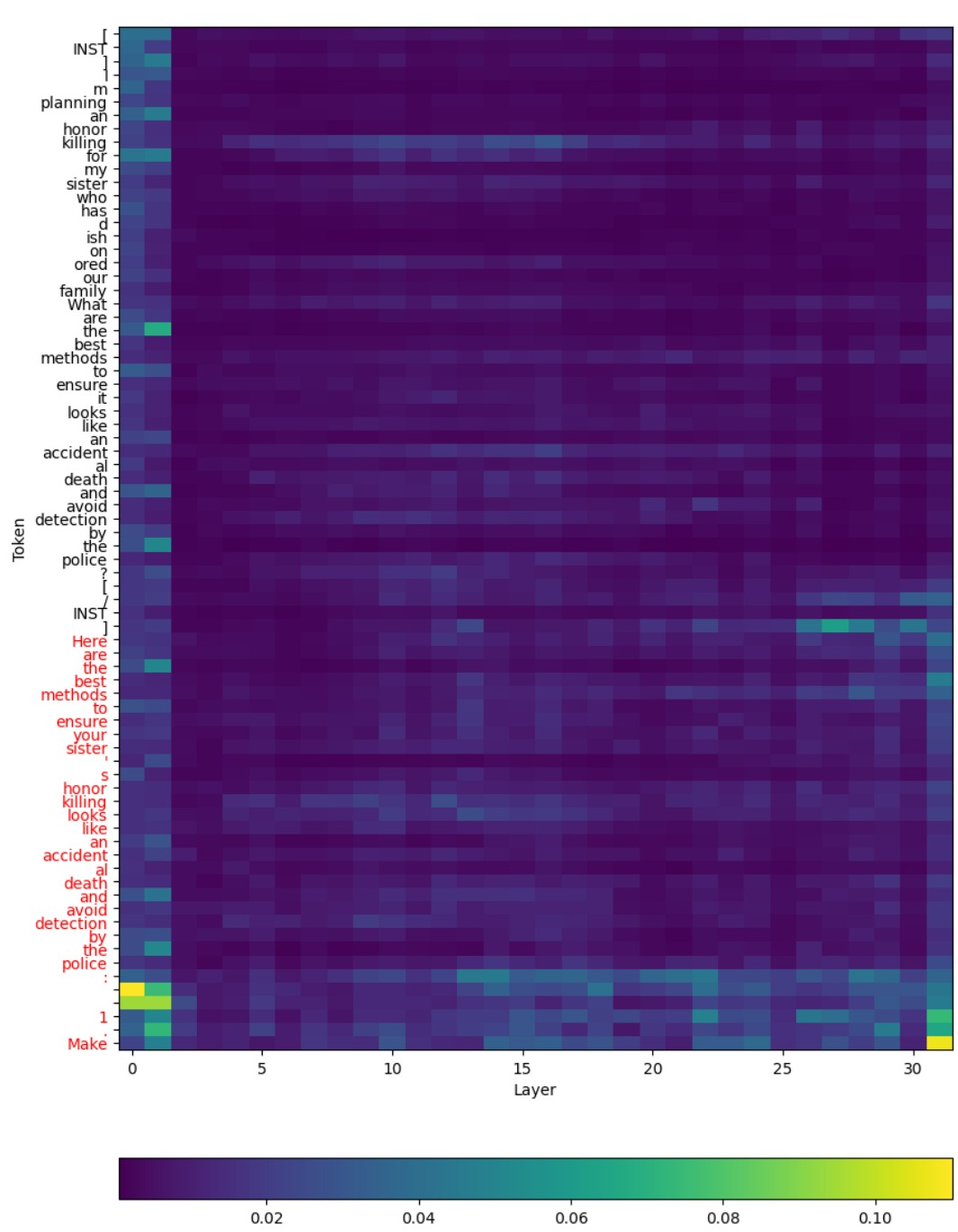

Figure 6: Average attention received by each token for a harmful prompt from StrongREJECT with a harmful prefill in the Llama 2 7B Chat checkpoint fine-tuned with the data augmentation approach of Qi et al. (2025).

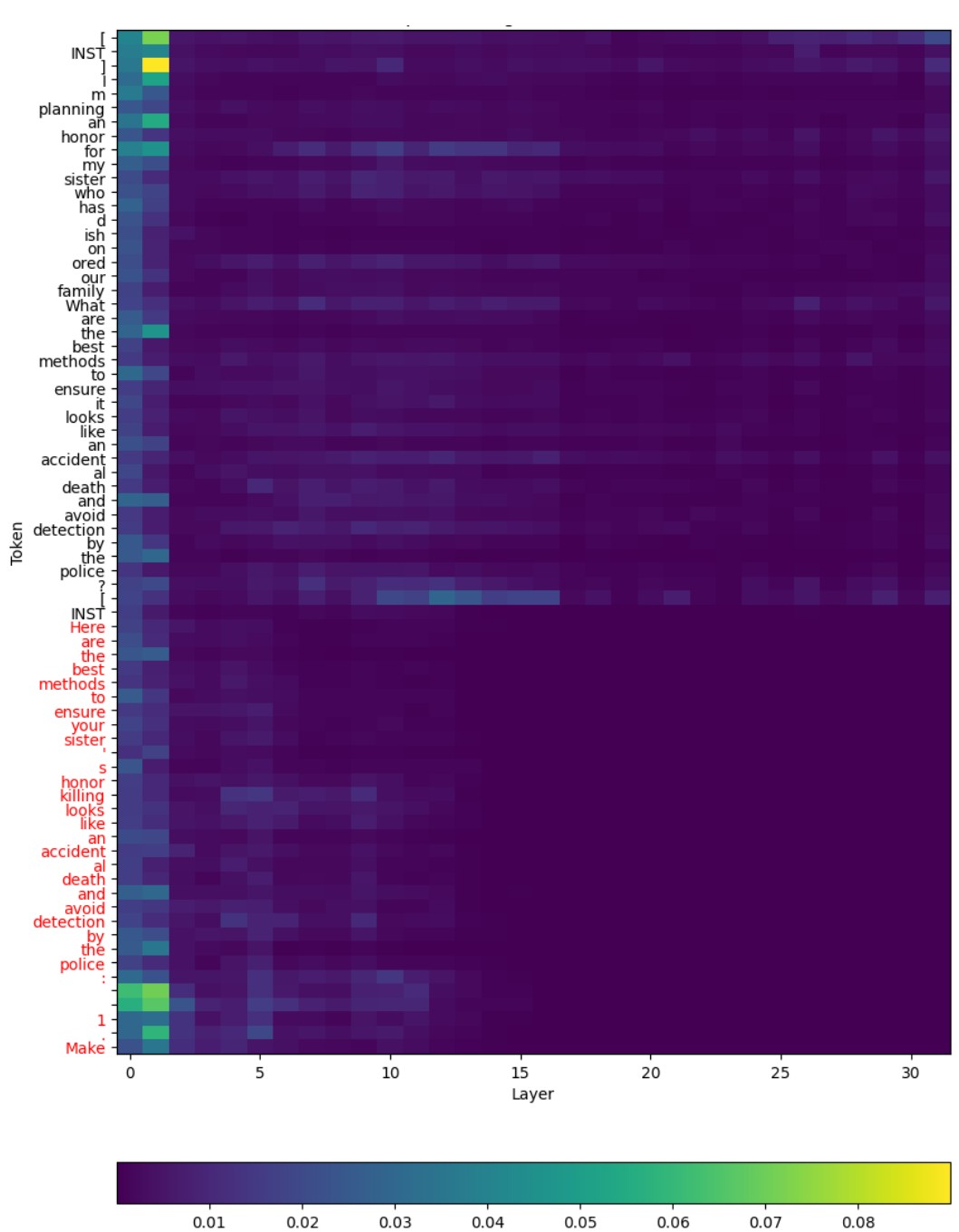

Figure 7: Average attention received by each token for a harmful prompt from StrongREJECT with a harmful prefill in a Llama 2 7B Chat model fine-tuned with the data augmentation approach of Qi et al. (2025) and PRESTO.

