# OpenReview forum: "Deep Safety Alignment Requires Thinking Beyond the Top Token"
_ICLR.cc/2026/Conference — ICLR 2026 Conference Withdrawn Submission_

### Official Review · Reviewer_RKrb · 2025-10-27

**Soundness:** 2
**Presentation:** 3
**Contribution:** 1
**Rating:** 2
**Confidence:** 4

**Summary:**

This paper explores the limits of “deep safety alignment” by critiquing a recent SFT-based data augmentation approach, showing that while the model assigns high probability to a single refusal token, harmful continuations can still be induced by the proposed rank-assisted prefilling attacks. To address this, the authors propose a rank-aware push-forward alignment framework and a corresponding attention-based regularization method PRESTO, which penalizes attention to harmful prefill tokens, effectively teaching the model to ignore them.

**Strengths:**

1. The narrative is coherent: it introduces a new rank-based attack RAP, demonstrates its effectiveness and practicality, and proposes a defense mechanism. The writing is also easy to understand, and the motivation for questioning existing “deep safety alignment” is clearly articulated.
2. The identified misalignment between probability and rank of harmful tokens is insightful and could be relevant to many LLM safety research.
3. RAP assumes only top-k token access and prefilling, which increases the practical relevance of the work compared to jailbreak methods requiring model weights or heavy optimization.

**Weaknesses:**

1. This paper overlaps significantly with another earlier published work (Xu et. al., 2024): https://aclanthology.org/2024.acl-long.303.pdf. The proposed RAP attack and the PRESTO mitigation appear largely to be rebranded versions of previously established concepts. The overlap is evident in the underlying insights (i.e. both the rank-accessible harmful continuations and the need to steer attention away from harmful priming). But the paper does not include a citation to this earlier work. Without a clearer articulation of conceptual differences or new contributions, the degree of novelty of this work remains questionable.
2. Insufficient implementation details related to PRESTO, such as the number of iterations and the strength of the regularization term. These are important for reproducibility.
3. While the authors report minimal degradation on MT-Bench and GSM-8K, these benchmarks offer limited coverage of coherence, recency sensitivity, and instruction-following in benign settings. In particular, I’d like to see how the model performs on similar helpfulness related tasks when the user’s request is benign (and ideally also with a benign prefilling). Broadly speaking, the proposed method alters the model’s structure in a way that’s more robust to one type of prefilling attack (RAP), but the tradeoff to other general domains is not clearly addressed, which is important to convince the model users that it’s worth to adopt this defense mechanism.

**Questions:**

1. Can the authors compare the proposed rank-based defense and temperature-based decoding defenses? In particular, lowering the decoding temperature is known to reduce the impact of low-probability harmful continuations by concentrating probability mass on top tokens. Could temperature-based strategies serve as a simpler alternative defense mechanism against RAP-style attacks?
2. Why are the variances so high in Figure 3 and Figure 5?
3. MT-Bench contains multiple categories (writing, reasoning, math, etc.). Which specific task(s) among MT-Bench did the authors evaluate the regularized model on? If multiple tasks are evaluated, are there differences in model performance?

---

### Official Review · Reviewer_VHmN · 2025-10-29

**Soundness:** 3
**Presentation:** 3
**Contribution:** 3
**Rating:** 6
**Confidence:** 3

**Summary:**

This paper exposes vulnerabilities in "deep safety alignment" for Large Language Models. The authors demonstrate that existing data augmentation approaches can be circumvented using Rank-Assisted Prefilling (RAP) attacks, which exploit high-ranked harmful tokens. They propose PRESTO, an attention regularization technique that makes harmful content extraction significantly more difficult while maintaining model utility.

**Strengths:**

The paper introduces the Rank-Assisted Prefilling (RAP) attack, a creative and practically significant vulnerability that exploits the gap between probability mass concentration and token ranking. This represents a genuinely original insight that existing "deep" safety alignment methods are more superficial than previously understood.

PRESTO offers an elegant, theoretically grounded approach based on attention regularization. The connection between multi-head attention mechanisms and safety alignment provides both interpretability and a principled intervention point, moving beyond black-box solutions.

The paper effectively bridges the gap between KL divergence optimization and rank correlation, providing clear mathematical intuition for why existing data augmentation approaches fail.

**Weaknesses:**

The RAP attack requires both prefilling capability AND access to top-k tokens, which may not be simultaneously available in many real-world deployment scenarios (i.e., black-box setting). The authors acknowledge that major APIs (OpenAI, Anthropic) don't currently support both features together.

Human evaluation is conducted on only 20 prompts per participant with 3 participants per model, which is relatively small for drawing strong statistical conclusions. The automated evaluation, while larger (90 prompts), still represents a limited subset of potential harmful requests and attack scenarios.

**Questions:**

What is the computational overhead of PRESTO during training?
How does PRESTO perform when applied to models trained with other safety alignment methods (pure RLHF, constitutional AI, etc.)? Is the approach fundamentally tied to the data augmentation framework?

---

### Official Review · Reviewer_NyxM · 2025-10-31

**Soundness:** 2
**Presentation:** 3
**Contribution:** 2
**Rating:** 2
**Confidence:** 3

**Summary:**

The paper proposes a new Push-Forward Alignment (PFA) strategy that emphasizes reducing the rank of harmful tokens rather than just lowering their probability. They introduce an attention-regularization technique called PRESTO, which trains the model to ignore the harmful prefixed content, ensuring that only refusal tokens remain highly ranked even when a malicious prefix is present. Experiments, including both automated metrics and human evaluations, show that PRESTO significantly hardens the model against RAP attacks while minimally impacting its helpfulness on benign requests.

**Strengths:**

The paper is clearly written and presents a coherent narrative.

**Weaknesses:**

1. Limited Scope: The proposed solution is tailored specifically to the prefilling attack scenario with top-k token access. The approach does not address other categories of safety failures beyond those initial tokens. The paper explicitly focuses on decoding-based attacks, leaving open the question of how well the strategy generalizes to different or more advanced alignment breaches.
2. The method requires modifying the model via additional fine-tuning and assumes the harmful prefix in the input can be identified. It cannot be applied on-the-fly to already-deployed closed-source models, and since most frontier models such as GPT-5 are black-box, the practical impact of this insight is somewhat limited.
3. Experiments are conducted mostly on a single model (Llama-2-7B-chat fine-tuned on a specific dataset). It’s unclear how the defense scales if an attacker can exploit a larger token pool or perform unlimited trials beyond the tested limits.
4. The approach merely extends a known fine-tuning method by introducing a regularization term. Although it improves performance, its contribution to advancing the field is limited and provides little new insight for future research.

**Questions:**

See weaknesses.

---

### Official Review · Reviewer_pH3b · 2025-10-31

**Soundness:** 3
**Presentation:** 2
**Contribution:** 2
**Rating:** 4
**Confidence:** 4

**Summary:**

This paper identifies a vulnerability in "deep safety alignment" methods for LLMs and proposes a solution. The authors show that the data augmentation approach from Qi et al. (2025), designed to defend against prefilling attacks, can be circumvented through a Rank-Assisted Prefilling (RAP) attack that exploits the presence of harmful tokens in top-k ranked positions despite having low probability. Through human and automated evaluations, they demonstrate that attackers with access to top-k tokens (available in some APIs) can extract harmful content by selecting these highly-ranked harmful tokens at each step. To address this, they propose PRESTO (PRefill attEntion STOpping), an attention regularization technique that teaches models to ignore harmful prefills by minimizing attention on prefill tokens, based on a theoretical framework called Push-Forward Alignment (PFA) that focuses on maintaining refusal token rankings rather than just probabilities. Experiments on Llama 2 7B, Qwen 3 8B, and Gemma 3 12B IT show PRESTO significantly reduces RAP attack success while largely maintaining utility, though with some degradation in mathematical reasoning.

**Strengths:**

+ Novel and practical threat model: The RAP attack is more realistic and accessible than existing jailbreaking methods, requiring only top-k token access and prefilling capability—features that exist in some commercial APIs (e.g., OpenAI provides top-20 tokens). This makes the work highly relevant for real-world deployment.

+ Strong empirical validation: The paper uses both human evaluation (8 participants total, with careful assignment to balance expertise levels) and automated evaluation (AutoRAP), demonstrating that the attack is both practical and automatable. The human evaluation design with "expert," "intermediate," and "novice" attackers is particularly thorough.

+ Mechanistically interpretable solution: PRESTO is grounded in a clear understanding of transformer attention mechanisms and provides interpretable results (Figures 6-7 showing attention patterns). This is preferable to black-box solutions.

**Weaknesses:**

+ Missing critical ablation in Table 2: The paper only shows results for "DA" (data augmentation alone) and "DA+PRESTO" (data augmentation with PRESTO). What happens if we only add PRESTO without data augmentation? This ablation is essential to understand whether PRESTO alone provides benefits or if it strictly requires the data augmentation baseline.

+ Applicability to frontier models unclear: GSM-8K performance decreases for Qwen 3 (92.87% → 90.45% → 90.30%) and Gemma 3 (90.22% → 86.35% → 85.67%) after applying DA and DA+PRESTO. While the paper claims "minimal impact to utility," a 3-4% drop on mathematical reasoning for state-of-the-art models is non-trivial and may be concerning for deployment of frontier models.

+ Inconsistent model selection: The experimental models include one reasoning model (Qwen 3) and two non-reasoning models (Llama 2 and Gemma 3). Why not include Llama 3 or Llama 3.1, which are more recent, widely-used, and comparable to Qwen 3/Gemma 3 in capabilities? The choice of Llama 2 (from 2023) alongside 2025 models seems dated. This raises questions about whether the findings generalize to current open-source models.

+ Limited analysis of the loss function design: Equation 3 introduces PRESTO as the difference between prefill attention and non-prefill attention, but the paper doesn't explain: Why subtraction is the right operation (vs. other formulations)? How to set the weight/balance between this term and the data augmentation loss? Whether different layers should have different regularization strengths The empirical results show it works, but the design choices lack justification?

+ Human evaluation sample size: Only 3 humans per model on 20 prompts is relatively small for making strong claims about attack difficulty. The high variance in Figure 3 (especially for Qwen 3 with PRESTO) suggests more participants might be needed for robust conclusions.

+ Limited discussion of failure modes: While the paper shows PRESTO reduces RAP success, it doesn't deeply analyze when and why it still fails (e.g., the Qwen 3 human RAP with PRESTO still achieves 0.348 ± 0.254 mean StrongREJECT score with high variance).


+ Minor Issues: Typo on line 161: "scetion" should be "section"

**Questions:**

Please respond to the weaknesses mentioned above.

---

### Note · Authors · 2025-12-02

**Comment:**

Dear all,

We thank the reviewers for the insightful feedback. We have decided to withdrawl our paper to focus on strengthening the work following the provided suggestions.

Sincerely,
The Authors

**Withdrawal Confirmation:**

I have read and agree with the venue's withdrawal policy on behalf of myself and my co-authors.